# Building Individual Player Performance Profiles According to Pre-Game Expectations and Goal Difference in Soccer

**DOI:** 10.3390/s24051700

**Published:** 2024-03-06

**Authors:** Arian Skoki, Boris Gašparović, Stefan Ivić, Jonatan Lerga, Ivan Štajduhar

**Affiliations:** 1Department of Computer Engineering, Faculty of Engineering, University of Rijeka, Vukovarska 58, 51000 Rijeka, Croatia; arian.skoki@riteh.uniri.hr (A.S.); boris.gasparovic@riteh.uniri.hr (B.G.); ivan.stajduhar@riteh.uniri.hr (I.Š.); 2Department of Fluid Mechanics, Faculty of Engineering, University of Rijeka, Vukovarska 58, 51000 Rijeka, Croatia; stefan.ivic@riteh.uniri.hr; 3Center for Artificial Intelligence and Cybersecurity, University of Rijeka, R. Matejcic 2, 51000 Rijeka, Croatia

**Keywords:** optimization, model fitting, energy expenditure, soccer, player profiling, GPS data

## Abstract

Soccer player performance is influenced by multiple unpredictable factors. During a game, score changes and pre-game expectations affect the effort exerted by players. This study used GPS wearable sensors to track players’ energy expenditure in 5-min intervals, alongside recording the goal timings and the win and lose probabilities from betting sites. A mathematical model was developed that considers pre-game expectations (e.g., favorite, non-favorite), endurance, and goal difference (GD) dynamics on player effort. Particle Swarm and Nelder–Mead optimization methods were used to construct these models, both consistently converging to similar cost function values. The model outperformed baselines relying solely on mean and median power per GD. This improvement is underscored by the mean absolute error (MAE) of 396.87±61.42 and root mean squared error (RMSE) of 520.69±88.66 achieved by our model, as opposed to the $B_{1}$ MAE of 429.04±84.87 and RMSE of 581.34±185.84, and $B_{2}$ MAE of 421.57±95.96 and RMSE of 613.47±300.11 observed across all players in the dataset. This research offers an enhancement to the current approaches for assessing players’ responses to contextual factors, particularly GD. By utilizing wearable data and contextual factors, the proposed methods have the potential to improve decision-making and deepen the understanding of individual player characteristics.

## 1. Introduction

During analysis of low-scoring team sports such as soccer, game status primarily falls into three categories: winning, losing, or drawing. The game status serves as a measure of performance achievements, significantly influencing a player’s level of effort [1]. Evaluating effort in soccer poses a challenge because of the sport’s complexity, with an average of approximately 1330 activities occurring during a game, involving activity changes every 4–6 s [2]. These activities, classified based on intensity zones, require both aerobic and anaerobic capacities. Evaluating effort is valuable for performance analysis, helping coaches and athletes identify strengths and weaknesses and optimize strategies. However, there are other situational variables, like game location, opponent level, and players’ fatigue, that can also affect soccer effort [3].

Sports journalists have been speculating a lot about the impact of the scoreline on a player’s work rate during a game. The commentators often mention that teams winning a game “sit on their leads”, while teams trailing “chase the game” [4,5]. Because of these ideas, academics have studied how the scoreline affects different parts of sports performance. Changes in a team and individual strategy, in reaction to the scoreline, demonstrate the importance of this situational variable. Section 1.1 will delve into diverse methodologies for evaluating players’ physiological responses to fluctuating scorelines and various contextual influences, as well as critically evaluate the limitations inherent in existing approaches.

### 1.1. Related Work

There are two primary approaches to measuring athletes’ physical performance: external load, which assesses the work they perform, and internal load, which evaluates how this work affects the individual [6]. In soccer, the gold standard for measuring external load is through GPS wearable devices [7]. These devices capture various metrics, such as total distance (TD) covered, number of accelerations and decelerations, distance covered in sprinting zones, and high-intensity actions [8]. However, a drawback of GPS devices is the requirement for athletes to wear them, necessitating careful attention from coaching staff [9].

With the advancement of video tracking technology and the installation of camera systems in stadiums, an alternative method has emerged. This technology extracts the players’ data points from the video footage, providing their positions on the pitch, usually at 25 frames per second [10]. Tracking data offers the same features as GPS trackers, with very close accuracy, but includes additional contextual details such as the positions of opponents and the ball, which can greatly aid tactical analysis [8]. However, implementing tracking data requires infrastructure, and GPS trackers offer more flexibility in day-to-day usage, especially when teams move between different training and match locations.

The internal load of soccer players can be measured through simple methods, such as filling out Rate of Perceived Exertion (RPE) questionnaires [11]. Through these surveys, players rate how hard a particular session was on a scale of 1 to 10 (category-ratio scale) or 6 to 20 (Borg scale) [12]. While this method is efficient and cost-effective, athletes need to be educated about its purpose and motivated to provide honest responses after each session. The subjectivity of the approach is a drawback, prompting consideration of more objective alternatives, like heart rate (HR) sensors, which provide valid insights about players’ internal states [13]. Enhanced athlete understanding typically correlates with greater answer certainty. However, the abundance of diverse data necessitates soccer coaches’ careful selection and prioritization of the most relevant and specific measures [14].

The rest of the section will explore various methodologies for utilizing both internal and external load data to assess player performance about different contextual factors and GD. Table 1 provides a comprehensive overview of research papers, which will be discussed later in the text.

The term “sitting on lead” indicates that winning status is a comfortable state for a team, which was observed in both top-class [15] and amateur soccer [16]. Contrary to that conclusion, other studies [17,18,19] claim that the TD covered is greater while winning, but Paul et al. [20] conclude that the game performance of elite soccer players is affected by a multitude of factors (such as fatigue, pacing, contextual, and tactical). These factors (e.g., possession) have a strong impact on performance. Teams with low possession have to cover more distance [21]. On the other hand, Barrera [22] concluded that the TD across each speed threshold is greatest when the team is drawing. A similar conclusion was reached by Redwood-Brown et al. [23]. They state that activity profiles change in a non-linear manner with changes in goal differences (GD).

The influence of the GD on the playing position’s physical performance was considered by several studies [5,24,25,26]. All of them agreed that the forward (FW) position covered significantly greater distances in winning games, while defenders showed the same trend for the losing games. A more recent study [26] supported the well-established finding that center midfielders (CMs) covered a longer TD than the other playing positions, with a trend of increasing activity in the winning game state.

Although many agree that scoreline influences game performance [5,15,16,17,18,19,22,23,24,25,26], Bloomfield et al. [27] found no significant impact of the scoreline or the interaction of scoreline and position on work rate. They concluded that midfield players tend to engage in more exercise than forwards, and that the intensity increases following scoreline changes are short-lived. O’Donoghue et al. [28] arrived at a similar conclusion, noting a decrease in V-cut path changes per minute following the first goal for both winning and losing teams. They also stated that there is no difference in work rate between teams achieving different outcomes.

**Table 1 sensors-24-01700-t001:** Categorization of related works. Arrows represent an increase, decrease, or maintenance of intensity depending on the game status (winning, drawing, losing). An asterisk (*) indicates that the intensity changes depending on the position or contextual factors. The position impact column gives information on whether research investigates playing position differences.

Authors	Winning	Drawing	Loosing	Position Impact
Lago et al. [15]	↗	-	↘	-
Shaw, Donoghue [16]	↗	-	↘	-
Moalla et al. [19]	↗	-	↘	-
Bordon et al. [17]	↘	-	↗	-
Castellano et al. [18]	↘	-	↗	-
Barrera [22]	-	↗	-	-
Redwood-Brown et al. [23]	-	↗	↗	-
Redwood-Brown et al. [5]	*	-	*	✓
Andrzejewski et al. [24]	*	-	*	✓
Bradley and Noakes [25]	*	-	*	✓
Lago-Penas et al. [26]	*	-	*	✓
Bloomfield et al. [27]	→	→	→	-
O’Donoghue et al. [28]	→	→	→	-

Several methodologies have been employed to evaluate how player performance fluctuates in response to GD and other contextual factors. However, divergent conclusions across studies underscore the necessity for a more tailored approach. While categorizing players by position enhances alignment in research, substantial discrepancies persist due to individual player profiles. In essence, the principal limitations of current studies include:Absence of an individualized approach to accommodate inter-player disparities;Lack of publicly available datasets for assessing and contrasting various methodologies; andInadequate mechanisms for operationalizing methods to provide coaching staff with novel player insights.

### 1.2. Contributions and Structure

In previous research, the primary emphasis has been on comparing metrics like TD, high-speed running (HSR) distance, or running and jogging distance across different GD scenarios, aiming to draw general conclusions about specific playing positions or leagues. However, given the unique nature of each team and individual player, a personalized approach is necessary to assess their effort accurately in various GD situations. Moreover, it is crucial to consider the contextual factors that influence a player’s performance, such as coaching instructions and tactical decisions, thereby requiring careful interpretation of the findings. While the existing papers have utilized different statistical analysis tests to draw conclusions, these approaches can be challenging to implement consistently in practice.

In this paper, we propose a mathematical approach utilizing optimization methods that consider individual player traits and specific distributions, departing from the prevalent group-centric approach in the current research. By employing two optimization algorithms, we evaluate the reliability of our approach, pinpointing potential areas of weakness. Thus, the primary contributions of this paper are as follows:Development of a mathematical framework employing optimization methods tailored to individual player traits;Provision of a publicly available dataset and source code for replication and further exploration; andImprove the validity of actionable player insights for more informed decision-making.

The structure of this paper unfolds as follows. Section 2 carefully explains the steps involved in data acquisition and preprocessing, detailing the procedures for constructing and assessing the mathematical model. Additionally, it outlines the optimization process for selecting parameters that most accurately encapsulate player attributes. The outcomes of the optimization algorithms across various players and matches, along with player profiling visualizations, are discussed in Section 3. Subsequently, Section 4 critically analyzes the obtained results, exploring their implications and potential interpretations. Finally, Section 5 summarizes the paper with concluding remarks and suggestions for future research.

## 2. Material and Methods

This section is structured into five distinct components, each addressing specific stages of constructing an energy consumption model. The initial part focuses on data acquisition, the second on data preprocessing, the third articulates the optimization objective, the fourth explains the optimization methods used, and the fifth section details the practical implementation of optimization for each player, alongside the expected outcomes. We offer a complete source code, including data and instructions, for reproducing the results and generating the plots. The code is available at https://github.com/askoki/expectation-score-profiler (accessed on 28 February 2024).

### 2.1. Study Design

Data collection occurred throughout the competitive season (2021–2022) of a professional soccer club that competed in a top-tier national division. Player exertion was quantified using GPS wearable sensors, specifically the GPexe pro^2^ (Exelio srl, Udine, Italy), with a sampling rate of 18 Hz. Before each match, probabilities of winning or losing were extracted from a betting website. Post-match data on the timing of goals scored and conceded were incorporated into the dataset. Players were categorized into five primary playing positions based on the coaches’ classification at season-end. These positions comprised: (1) center back (CB) for central defending, (2) wide back (WB) for wide defending, (3) midfielder (MF) for central midfield, (4) forward (FW) for central attacking, and (5) wing forward (WF) for wide attacking roles.

### 2.2. Subjects

The dataset comprised 33 games, played by 19 male soccer players (age 25±3 years; height 180±6 cm; weight 75±6 kg). Throughout these games, 3135 min of game time were recorded. Player distribution across playing positions was as follows: 5 CBs, 5 MFs, 5 FWs, 3 WFs, and 1 WB. The entire analysis was made on the same team without revealing the player’s and clubs’ identities. The data from the opposition was not available for analysis. This research was conducted under the guidelines outlined by the Ethical Committee of the University of Rijeka (REF: 2170-1-43-29-23-2), in compliance with the principles of the Declaration of Helsinki.

### 2.3. Data Acquisition

Players wore lightweight vests equipped with small devices on their backs for convenient monitoring. Ten minutes before each game, players would put on the GPexe pro^2^ sensors, which provided sufficient time for calibration. These wearables were equipped with: GPS, gyroscope, accelerometer and magnetometer sensors. Post-game, data was retrieved from the sensors and segmented into 5-min intervals, chosen as the optimal balance between post-processing efficiency and granularity of recorded data. While sensors can extract various features, this paper focuses only on energy expenditure, which was measured in J/kg. Detailed information on energy calculation in the GPexe system is thoroughly described in a paper by di Prampero et al. from 2005, determining the energy cost of acceleration on flat terrain [29]. To better illustrate the 5-min interval energy expenditure, an example of a single game and one player is shown in Figure 1. The energy parameter takes into account both the energy required to cover the given distance (at constant speed), as well as the energy needed to perform speed variations. The latter is calculated by equating accelerated running and sprinting to uphill constant running, which was described by Osgnach et al. [30].

The other crucial part of the data comprised the win, draw, and lose ratios for each game played by the observed team, which was sourced from the website http://clubelo.com (accessed on 6 November 2023). The odds displayed on the website reflected the probabilities of winning or losing before each championship game. The third component involved scoring and conceding minutes for the observed team throughout the entire competitive season. These timings were extracted from the official competition website after the conclusion of each game.

### 2.4. Data Preprocessing

For each fixture, an expectation *e* was calculated by dividing the probability of winning a game by the probability of losing it. If the resulting *e* was equal to or greater than 2, the team was considered the favorite, while if it was less than 0.5, the team was deemed the underdog. Games falling within the *e* range of 0.5 to 2 were classified as closely contested games. The dataset included 21 games where the team was the favorite to win, 13 close games, and only 2 games in which the team was considered the underdog. Given the scarcity of games featuring the team as the underdog, the analysis was simplified into states of being either the favorite or non-favorite.

The influence of the scoreline on the energy expenditure was limited to GD of −2, −1, 0, 1, and 2. GD refers to a difference in the score of the team analyzed in this paper versus the opponent they are playing against. The border values (−2 and 2) included all the situations where the team was losing by 2 goals or more or was winning by 2 goals or more, respectively. The resulting distribution of game time concerning a particular GD can be seen in Figure 2. The most dominant GD was 0 because the game starts at 0-0, and this scoreline is the most frequent one in the full 90 min of the game. Drawing minutes that differed from the starting 0-0 score included only 272 min (out of 1401, i.e., 19.4%). GD distribution is asymmetric and negatively skewed (left-skewed, shifting mean to the right/positive score). This is likely because the observed team was one of the top teams in the league and a favorite to win in a lot of games (21 out of 36).

The player energy expenditure dataset included information concerning the start and end timestamps. This was combined with the scoreline dataset to join GD information with energy expenditure. The values close to the start of the game and the halftime—when players were waiting in the dressing room—were discarded from the evaluation.

### 2.5. Energy Expenditure Model

As mentioned in Section 2.3, the data concerning a player’s energy is collected by the sensors in 5-min periods. In general, energy consumption is assumed to be influenced by the current game outcome, i.e., a win, draw or loss, as well as contextual factors, such as whether a team is the favorite or underdog or whether it is a tightly-contested game. It is important to understand that the values for energy expenditure can vary from player to player. In addition, the effect of fatigue on a player’s performance, especially in the latter stages of the game, is an important factor that should not be overlooked. The hypothesis is that a player’s work rate changes depending on the GD, expectation and ability to maintain intensity, i.e., endurance. We have created a model that captures each player’s performance characteristics, based on the data compiled from the games they have played.

For game *j*, we define *e* as one of two categories: favorite (f) or non-favorite (nf), which can be represented as $e_{j}\in \left\{f,nf\right\}$. Furthermore, we consider GD as a function of game time *t*, where GD can take values from the set $d_{j}\left(t\right)\in \left\{-2,-1,0,1,2\right\}$, which is considered as a known input to the energy expenditure model.

To parameterize players’ performance, we defined score performance parameters $P_{d,i}$ for player *i*. These parameters remain constant across all analyzed games during the season and need to be estimated from the available data. Given our assumption that the performance of each player evolves not only with the score but also with pre-game expectations, we model the current power exerted by player *i* in game *j* as a function of both time and expectation, denoted as $P_{i}\left(d_{j}\left(t\right),e_{j}\right)$.

Finally, the energy of each player spent during the game, $E_{i,j}\left(t\right)$, is regulated using a simple differential equation, with a starting condition $E_{i,j}\left(0\right)=0$:(1)$$\frac{dE_{i,j}}{d\tau }=\sum_{\tau =0}^{\tau =t_{e,i,j}-t_{s,i,j}}P_{i}\left(d_{j}\left(\tau +t_{s,i,j}\right),e_{j}\right)\cdot \tau \cdot \eta_{i}\left(\tau \right).$$

Here, we denote η as the power consumption efficiency, represented by a decaying exponential function $\eta_{i}\left(\tau \right)=e^{-\alpha_{i}\cdot \tau }$. The parameter $\alpha_{i}$ is determined as $-\ln\eta_{90}/90$.

In this context, $\eta_{90}$ represents the coefficient of the player’s energy tank at the 90 min of playing time, α is a coefficient describing the player’s condition, and τ is the individual player’s playing time, which can differ from game time depending on whether a substitute player was involved or not. If we define $t_{s,i,j}$ as a game *j* time when player *i* entered the field, then $\tau_{i,j}=t-t_{s,i,j}$. Similarly, $t_{e,i,j}$ defines the time when the player exited the game. It is trivial to show playing time in the game’s absolute time: $t\in [t_{s,i,j},t_{e,i,j}]$, as used for expended energy and GD functions, or in player’s relative time (playing time) $\tau \in [0,t_{e,i,j}-t_{s,i,j}]$, as used in power consummation efficiency function. Energy expenditure model (Equation 1) is solved using the Euler method with Δt=1min, for the game time interval $t\in [0,t_{end,j}]$, where $t_{end,j}$ is the total duration of game *j*.

Figure 3 illustrates an sample solution of the energy model presented in Equation (Equation 1). It explains the dynamics of energy expenditure, taking into account player-specific characteristics such as pre-game expectations (*e*), physical and motivational parameters $P_{d,i}$, endurance $\eta_{i}$ and the evolving score $d_{j}\left(t\right)$. Key assumptions for constructing this sample solution include (1) a player’s participation in the full 90 min of a game, (2) the fluctuation of GD as shown in the middle graph, (3) the power used corresponding to GD as shown in the bottom graph, (4) the fact that the player’s team is the favorite to win, and (5) a $\eta_{90}$ coefficient of 0.7. This method was used to derive the performance parameters and evaluate the endurance coefficient for all players. The detailed outcomes of this analysis are presented in the Section 3.

### 2.6. Model Fitting via Optimization

The model described in the previous section has some parameters that are unknown and need to be estimated. The resulting energy curve over the players’ playing time should come as close as possible to the real measurements of the GPS sensor. It is therefore obvious that we have to set up and solve this as an optimization problem.

Table 2 contains the list of all parameters that were used as optimization variables. The constraints for the power zones ($P_{i}$) were limited to values in the range $200\leq P_{i}\leq 800$. This is a valid range in which the player uses a predefined power to generate energy over a certain period of time—in this case, 5 min. The function influencing endurance, referred to as η, was calculated using a common approach to energy loss that uses a constant parameter, α. This parameter determines the rate of power loss over time. The calculation of the constant α is described in detail in Section 2.5, where the constraint on the parameter $\eta_{90}$ ensures that it falls within the valid range of 0.5 to 1.0. This range reflects the typical decrease in player intensity towards the end of a 90-min game. In most cases, players are clearly exhausted at the 90-min mark. A value of 1.0 means that the player can maintain the same intensity as at the beginning of the game, while the lower limit of 0.5 means that the player’s performance has decreased by 50% compared to the beginning of the game. All optimization variables are summarized in an optimization vector, which is shown in Equation (Equation 2). The cost function ε was calculated by minimizing the squared difference between the estimated and actual energy expenditure. This value was then divided by the number of minutes the player had played (Equation (Equation 3)):(2)$$\begin{array}{cc} x= & \left(\begin{array}{c} P_{-2,f},P_{-1,f},P_{0,f},P_{1,f},P_{2,f},\\ P_{-2,n},P_{-1,n},P_{0,n},P_{1,n},P_{2,n},\\ \eta_{90} \end{array}\right) \end{array},$$
(3)$$\varepsilon_{i}\left(x\right)=\frac{\sum_{j=1}^{j=n}\int_{t_{s,j}}^{t_{e,j}}{\left(E_{i,j}\left(x,t\right)-E_{measured,i,j}\left(t\right)\right)}^{2}dt}{\sum_{j=1}^{j=n}\left(t_{e,i,j}-t_{s,i,j}\right)},$$
where *n* is the number of recorded games, $t_{s}$ is the time in a game the player has entered the field, and $t_{e}$ is the time when the player left the field. Now, we can formulate the optimization problem whose solution represents the best-fitted parameters for an individual player: $\hat{x}={arg min}_{x}\varepsilon \left(x\right)$.

Model fitting is performed using two optimization methods: deterministic Nelder–Mead (NM) and stochastic particle swarm optimization (PSO). NM is a commonly used numerical method to find the minimum or maximum of a cost function in multidimensional space. It can be effective in optimizing functions with a small number of variables, but may have problems with high-dimensional problems or functions with non-smooth landscapes. On the other hand, PSO is a population-based metaheuristic algorithm that can handle a wide range of optimization problems, including those with high-dimensional spaces and nonlinear landscapes. However, PSO can prematurely converge to suboptimal solutions and is sensitive to the choice of parameters. The results of these algorithms provide insight into the stability of the proposed method for determining player motivation with respect to the physical parameters. A large deviation in the results would indicate that the problem is very challenging, which means that it is very difficult to find the optimum. The methods were implemented using the Python packages SciPy (https://docs.scipy.org/doc/scipy/reference/optimize.minimize-neldermead.html, accessed on 6 November 2023) and Indago (https://pypi.org/project/Indago/, accessed on 6 November 2023).

### 2.7. Procedures

Optimization of energy expenditure considering pre-game expectation and fatigue effect was performed for each player across all games played. Each optimization was repeated 10 times for each individual player. Every run yielded: (1) cost function change through evaluations, (2) parameter convergence of the best-performing run, (3) the resulting NM vectors, (4) the resulting PSO vectors, (5) the minutes per GD, and (6) the individualized GD influence matrix *g*. This matrix represents the positive or negative influence of GD and expectation. It is calculated in the post-processing phase using Equation (Equation 4) and first determines a reference parameter *r* for each GD scenario *d*. The parameter *g* is then computed by dividing the cumulative energy per GD (product of power and time) and expectation by the time spent in a particular GD. The resulting values range from 0.5 to 1.5 and describe the negative or positive impact of GD on a player’s energy expenditure.
(4)$$g_{e,d}=\frac{\sum_{j|d,e}P_{d,e}\cdot t_{d,e}}{r_{d}},\;r_{d}=\frac{\sum_{j|d}P_{d}\cdot t_{d}}{\sum_{j|d}t_{d}}$$

Depending on the resulting vectors of power and the endurance coefficient, the calculated energy expenditure was compared with the real one. This enabled detailed inspection of the algorithm quality through the game samples. An aggregated square of error was calculated for each player and game, separately for each method. Also, two baseline scenarios were introduced. The first baseline, denoted as $B_{1}$, utilized mean power values per GD, with η set to 1. The second baseline, referred to as $B_{2}$, employed median power values per GD, also with η set to 1. Model estimations were further compared to $B_{1}$ and $B_{2}$ by calculating standard regression metrics, including mean squared error (MSE), root mean squared error (RMSE), and mean absolute error (MAE). This comparison allowed for a thorough assessment of our approach’s performance relative to simpler methods.

## 3. Results

In this section, the performance and convergence speed of the PSO and NM algorithms are displayed, alongside a comparison with the baseline models, denoted as $B_{1}$ and $B_{2}$, for all observed players. Next, the results for a selected player are presented, along with an explanation of their utility and the validation process for understanding the player profile. Finally, the same player is used to illustrate the performance of the approach within the game in a specific context.

### 3.1. Model Performance

As noted in Section 2.6, the cost function was calculated by minimizing the squared difference between the estimated and the real energy expenditure. Table 3 displays the results of the PSO and NM optimization methods, in addition to the two baseline approaches, denoted as $B_{1}$ and $B_{2}$. It is evident that both PSO and NM consistently outperform the two baseline methods across all players.

Furthermore, it is worth noting that PSO and NM demonstrate convergence towards nearly identical cost function values for all players, with NM being slightly better. This suggests consistent and reliable results from both algorithms. The primary distinction lies in the number of evaluations required by PSO compared to NM. Typically, PSO takes a higher number of evaluations to reach optimal values compared to NM. Nevertheless, it is crucial to utilize both methods to assess the confidence level of each resultant parameter and ensure the validity of the information before potentially presenting it to the coaching staff.

To thoroughly evaluate the performance of the proposed approach, we calculated standard regression metrics for the model outlined in this paper and two baseline approaches, denoted as $B_{1}$ and $B_{2}$. For comparison, we have included a slightly more effective method, NM, to represent the model presented in this paper. Table 4 shows the mean and standard deviation of these metrics across all players utilized in the study, including MAE, RMSE and MSE. The proposed model outperforms the baseline approaches on all metrics, has a lower standard deviation between players and thus provides more stable results.

More detailed improvements across all players are provided in Appendix A, while the stability of the optimization results is elaborated in Appendix A, available in the Appendix A.

### 3.2. Individual Player Profile

Each player has a distinct and unique profile, so it is beneficial to analyze them individually. As described in Section 2.5, our optimization goal is to determine the power vector *P* with respect to the state of GD and the expectation *e*, taking into account the fatigue index η. The optimization results for a single player provide information about the changing performance level with respect to *e*, the current context of the game (e.g., GD) and the energy loss over the 90-min game duration. Figure 4 shows the most successful optimization run for the randomly selected *athlete3* playing in the FW position. The figure illustrates how GD and *e* influence the player’s physical effort, along with the endurance coefficient, which indicates his ability to sustain a given effort. This visualization facilitates practitioners’ understanding of individual changes in response to contextual factors. Each run provides the following results: (1) the change in the cost function through the evaluations (Figure 5a), (2) the parameter convergence of the best iteration (Figure 5b), (3) a radar plot representing the resulting vectors using the NM method (Figure 4a), (4) radar plot showing the resulting vectors using the PSO method (Figure 4b), (5) minutes per GD (Figure 6) and (6) individualized GD influence matrix *g* (Figure 7). The presented results illustrate how the players’ effort varies with GD, *e* and their respective endurance coefficient. In Section 3.3 we will explain how these results correspond to real-world data. We will also examine how the model’s estimates match different games and contextual scenarios.

### 3.3. Individual Game Analysis

To better understand how the proposed approach fits the game data, the three-game example is shown in Figure 8. In this analysis, we continue to focus on the same player introduced in the preceding section, namely *athlete3*, who plays in the FW position. These three selected games also cover different scenarios, namely: winning and drawing while being non-favorite, and losing a game while being a favorite.

In the first game (G1), the team was not considered the favorite to win, but they actually won the game. The calculated curve fits the data very well. The second game (G24) illustrates a scenario in which the team was also not considered the favorite to win, but ultimately drew. In this case, the model tends to underestimate the actual energy output, with the score fluctuating between GD−1 and GD1 before ending in a draw with GD0. The last game (G27) represents a case where the team was the favorite to win, but ended up losing the game. The curve closely matches the measured data, but overestimates the period from the 40th to the 60th minute. During this time, the opposing team scored another goal, shifting the GD to −2. Figure 9 provides a condensed overview of the calculated and actual values for the first ten games of the top five players, sorted by the number of games played. Comprehensive data on all other players and games can be found in Appendix A.

## 4. Discussion

A workload tied to GD differs from player to player, also when generalized across the positions in the team. Every playing position in soccer requires different physical characteristics. FWs and wide players tend to have more high-intensity actions that are shorter in duration. MFs usually cover the longer distances, and CBs have the most tackles and aerial duels but cover the least distance. As already mentioned in the Introduction, FWs tend to work harder when the team has the lead and the CBs when the team is losing.

Previous research in this field has predominantly utilized statistical methods to analyze responses across different teams, leagues, or playing positions. However, the applicability of these approaches is limited due to variations in data distribution and the challenge of transferring methods from one team to another. In contrast, our proposed approach utilizes modeling with exponential decay, offering enhanced flexibility in data fitting for more realistic outcomes. This adaptable model can be easily applied to any team and individual. Additionally, our approach considers variations in player behavior, accounting for pre-game expectations and changes in the GD. The concept of endurance, crucial to understanding the impact of player fatigue, is represented using exponential decay. This concept was first introduced in a study by Bartow et al. in 1991, where they described the relationship between oxygen uptake ($VO_{2}$) and heavy exercise [31]. Subsequent research has observed similar results in the context of repeated running sprints [32] and as an index of physical work capacity [33]. Furthermore, more recent studies have utilized exponential decay to determine the anaerobic power reserve in cyclists [34] and to understand the physiological demands and recovery kinetics in women’s soccer [35].

To validate our approach, we conducted a comparative analysis of the performance of PSO and NM optimized models against the baseline models, $B_{1}$, which takes the mean power per GD, and $B_{2}$, which uses the median of power per GD. The results showed that both PSO and NM methods consistently outperformed the baseline models when applied to the observed dataset, reaching similar optimal values. This represents a pioneering approach, being the first to model player energy expenditure based on both GD and exponential decay. To enhance our approach, we can delve deeper by considering changes in GD, such as GD−1 to GD0, GD0 to GD1, etc. Additionally, incorporating extended pre-game expectations (underdog, close game, favorite) has the potential to refine and adapt the model further. However, for a comprehensive analysis, this should be applied across the entire league and all matches to account for various possibilities. Gathering more data and comparing player profiles across different leagues would provide a basis for a thorough validation of this approach.

Furthermore, we examined how the model performed when compared to real game data, taking into account the dynamic and unpredictable nature of the game. While the model exhibited a good fit to the measured values, there were some outliers. These outliers could be attributed to unaccounted contextual factors, such as the ball’s position, the specific area of the pitch, the game tempo, and others. These factors represent promising avenues for future research expansion.

Constructing distinctive player profiles by integrating pre-game expectations, GD scenarios, and the endurance coefficient can offer soccer coaches fresh perspectives or confirm their existing insights about their players. It is evident that the volume of data associated with a player significantly impacts the resulting values, rendering them more dependable with larger datasets. The limitation of our study is that it was performed on only one team, and the additional validity needs to be checked by analyzing the results of other teams in different leagues.

It is crucial to recognize that a player’s effort is influenced not only by the variables addressed in this paper, but also by additional ones, such as the game context and specific tactical instructions from coaches. These factors should be taken into account by the coaching staff when interpreting the results and making informed decisions based on model profiling.

## 5. Conclusions

In summary, this paper introduces a methodology that utilizes real energy expenditure data collected through 5-min periods in soccer games to evaluate player effort. Our novel mathematical model, incorporating GD states and exponential decay, surpasses baseline approaches relying solely on mean and median values. Our approach’s primary objective is to objectively quantify players’ physical states, which fluctuate according to contextual factors and the mental profiles of players, offering valuable insights into their performance. Notably, this study is the first of its kind to combine optimization methods and GPS data to assess effort in soccer.

However, the practitioners should recognize that the resulting player profiles are not ideal, as they do not encompass the game context and tactical instructions, which also play a significant role in influencing energy expenditure. It is still a valuable tool for interpreting the results of the energy expenditure analysis and making informed decisions to optimize player performance.

Overall, our study underscores the need for a tailored and adaptable approach when analyzing energy expenditure and endurance in soccer, emphasizing the importance of considering individual player characteristics and an adaptive approach.

## Figures and Tables

**Figure 1 sensors-24-01700-f001:**
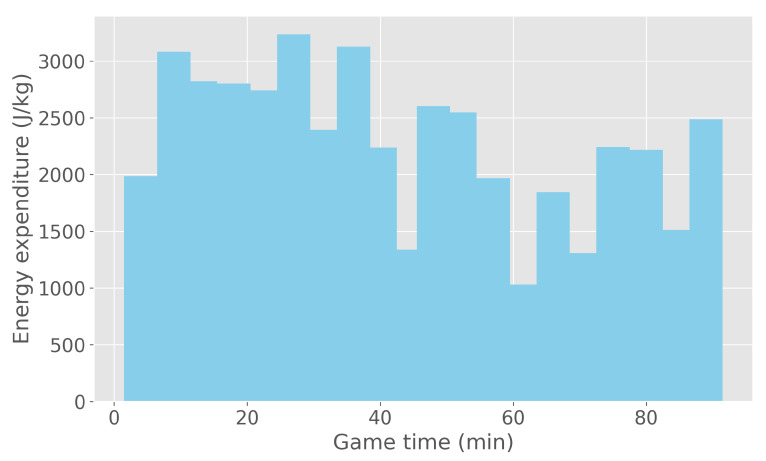
A randomly selected example of a 5-min interval energy expenditure in a single game for one player.

**Figure 2 sensors-24-01700-f002:**
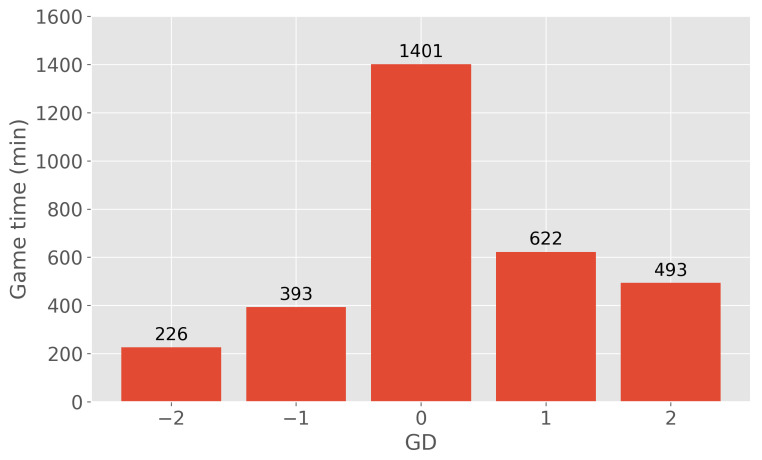
Distribution of game time with respect to the particular GD. The plot was generated by using all the available games and combining them with scoring minutes. A player would have the same distribution (as shown) if he participated in all the games and always played the full length of the game. GD refers to a difference in the score of the team analyzed in this paper versus the opponent they are playing against. The border values (−2 and 2) included all the situations where the team was losing by 2 goals or more or was winning by 2 goals or more, respectively.

**Figure 3 sensors-24-01700-f003:**
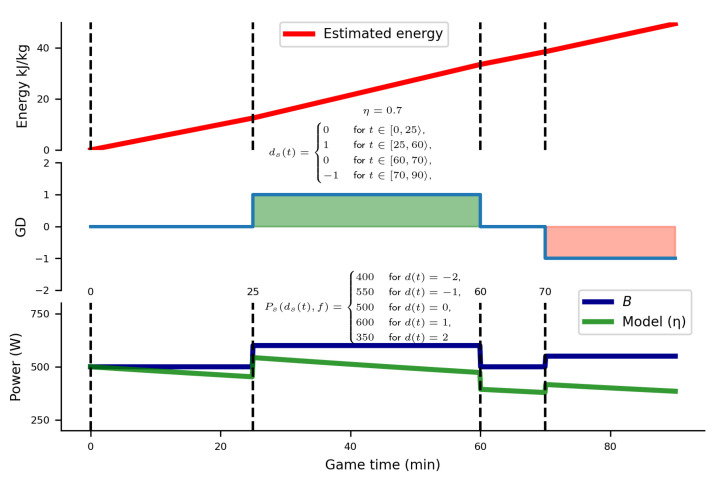
Model results for the sample use case. The player played 90 min with η being 0.7, and the team being the favorite to win. The remaining parameters, including power vector $P_{s}$ and GD change through the game $d_{s}\left(t\right)$, are depicted in the plot. The top plot showcases calculated energy expenditure values, accounting for both the impact of exponential decay with η and the variation in power per GD. The middle plot depicts the fluctuation of GD through the 90-min game, with positive GD values in green and negative GD values in red. The bottom plot represents the power associated with GD. The blue line illustrates power without fatigue (η) effects, while the green line highlights the influence of exponential decay (η) on the defined power per GD. Additionally, dotted vertical lines are incorporated to emphasize changes in GD throughout the game.

**Figure 4 sensors-24-01700-f004:**
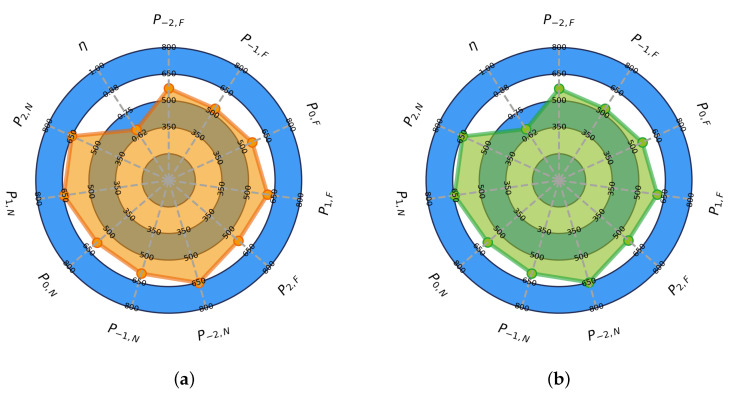
A depiction of the optimization results for *athlete3* playing in a FW position. The reliability of the obtained values can be assessed by examining the data density GD, as depicted in Figure 6. (**a**) NM method results. (**b**) PSO method results.

**Figure 5 sensors-24-01700-f005:**
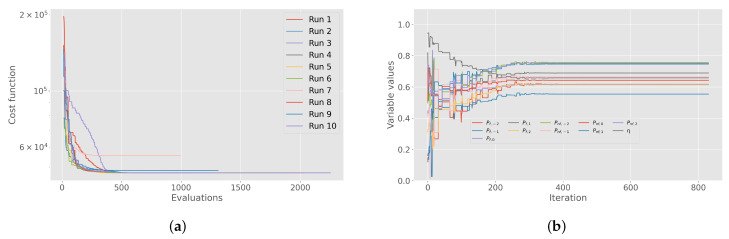
Convergence analysis of the PSO optimizer for *athlete3*. The left panel illustrates the convergence of the cost function through the iterations, with the Y-axis indicating cost function values on a logarithmic scale. On the right, the convergence of optimization variables during the best run is presented. Variable values were normalized with Min–Max scaling by using the constraints of 200≤P≤800 for power per GD and 0.5≤η≤1.0 for endurance coefficient. (**a**) Cost function values through iterations. (**b**) Best run variable convergence.

**Figure 6 sensors-24-01700-f006:**
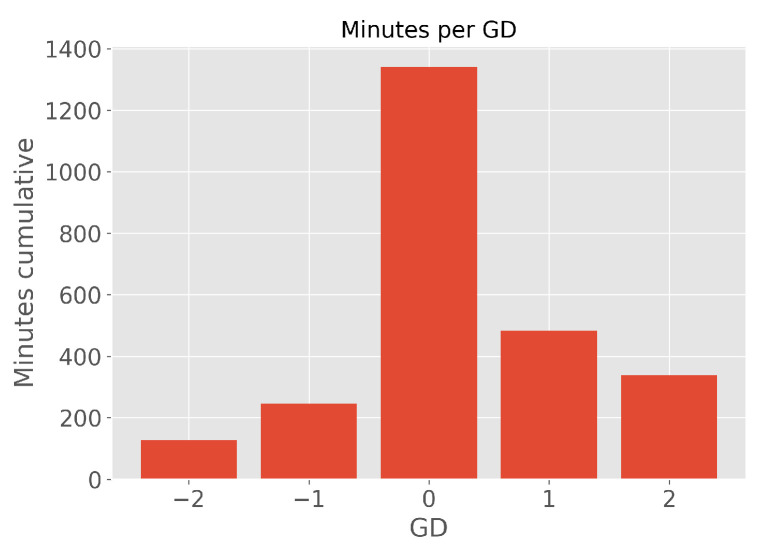
The number of minutes *athlete3* played per GD.

**Figure 7 sensors-24-01700-f007:**
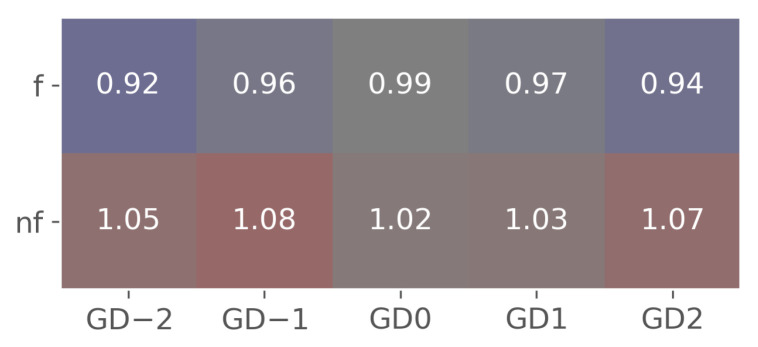
Influence matrix *g* for *athlete3*. Values above 1.0 denote an increase in effort compared to baseline state of the observed GD while values below indicate decrease of effort. The y-axis is labeled ‘f’ for favorite to win and ‘nf’ for non-favorite, while the x-axis represents GD ranging from −2 to 2.

**Figure 8 sensors-24-01700-f008:**
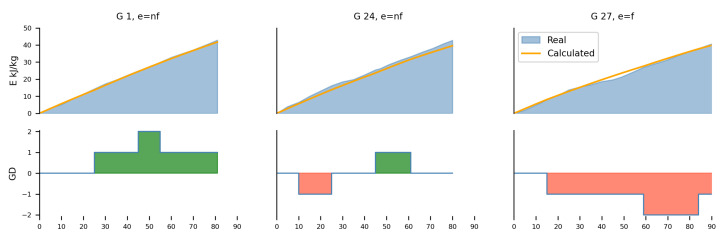
Three distinct gaming scenarios are illustrated: one where the team is non-favorite but wins a game, another where they are non-favorite but end up drawing the game, and the third where the team was favorite and loses the game. The upper graph illustrates the comparison between actual (blue) and calculated (orange) energy expenditure. The bottom graph shows the dynamic changes in GD throughout the match. The green color indicates positive changes in GD, while red color highlights negative changes in GD.

**Figure 9 sensors-24-01700-f009:**
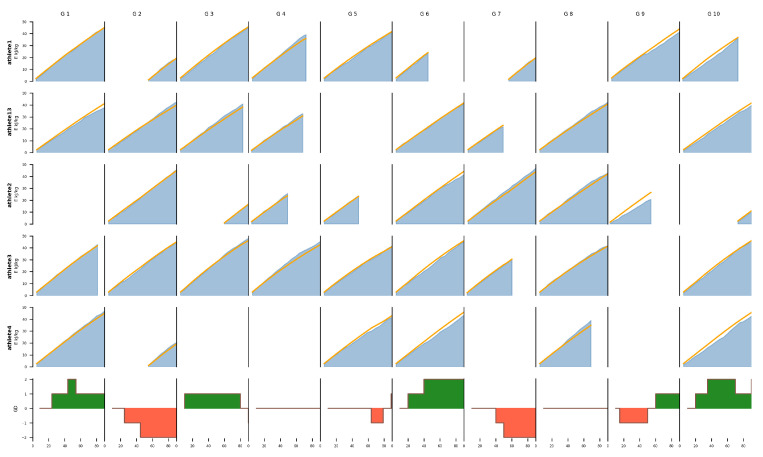
Overview of the top five players, ranked by the number of games played, across their first ten games. Each column represents an individual game (G1–G10), while rows correspond to players (Athlete1–4, Athlete13), with the last row indicating GD across 90 min for each observed game. The negative GD values are highlighted in red, positive values are in green. The orange line reflects calculated values using the proposed model, and the blue area illustrates real values measured via sensors.

**Table 2 sensors-24-01700-t002:** Overview of the optimization parameters. The variables of the power vector *P* were restricted to values in the range of 200≤P≤800, and the endurance parameter $\eta_{90}$ was constrained to a value between 0.5 and 1.0, which describes a valid range of the intensity of drop-in players after 90 min of play.

Parameter	Description
$P_{-2,f}$	Power if the team is the favorite to win and the GD is −2.
$P_{-1,f}$	Power if the team is the favorite to win and the GD is −1.
$P_{0,f}$	Power if the team is the favorite to win and the GD is 0.
$P_{1,f}$	Power if the team is the favorite to win and the GD is 1.
$P_{2,f}$	Power if the team is the favorite to win and the GD is 2.
$P_{-2,n}$	Power if the team is not the favorite to win and GD the is −2.
$P_{-1,n}$	Power if the team is not the favorite to win and GD the is −1.
$P_{0,n}$	Power if the team is not the favorite to win and the GD is 0.
$P_{1,n}$	Power if the team is not the favorite to win and the GD is 1.
$P_{2,n}$	Power if the team is not the favorite to win and the GD is 2.
$\eta_{90}$	Energy potential at the 90 min of the game.

**Table 3 sensors-24-01700-t003:** Comparison of PSO and NM number of evaluations and cost function error per player (calculated using Equation (Equation 3)). Both algorithms produce similar results; however, the number of evaluations required for converging in NM is much lower than in PSO. Numbers printed in boldface represent the best performing approaches with the lowest error.

Player	PSO Evals	NM Evals	$\epsilon_{\mathrm{PSO}}$	$\epsilon_{\mathrm{NM}}$	$\epsilon_{B_{1}}$	$\epsilon_{B_{2}}$
athlete1	4543	2640	51,216.06154	**51,215.88103**	58,570.7	59,138.72
athlete2	4257	2999	54,454.21483	**54,453.24721**	56,773.98	57,262.04
athlete3	3498	1787	47,458.30669	**47,453.25968**	60,483.51	60,915.34
athlete4	4257	2340	61,689.72877	**61,688.99921**	67,441.84	67,568.55
athlete5	5511	2267	51,176.87398	**51,176.17927**	53,610.1	56,329.67
athlete6	4257	1996	86,327.95668	**86,326.31882**	102,836.27	106,435.89
athlete7	3861	2413	68,392.25968	**68,381.8828**	71,460.11	73,394.38
athlete8	3971	1987	38,786.39576	**38,781.8148**	42,469.21	43,583.95
athlete9	4631	2211	55,862.58061	**55,859.77039**	69,456.67	68,891.53
athlete10	9713	2252	52,416.0806	**52,413.48422**	60,658.66	60,922.99
athlete11	3146	1822	112,254.95409	**112,249.95746**	354,583.69	711,259.76
athlete12	1892	1495	**53,581.03919**	**53,581.03919**	59,896.14	62,422.84
athlete13	11231	2317	42,831.99707	**42,830.85899**	46,532.61	46,725.41
athlete14	3465	2267	70,119.45093	**70,118.20461**	73,136.68	77,705.22
athlete15	5808	2376	46,316.69107	**46,316.25094**	52,382.71	53,579.38
athlete16	2849	1875	27,232.6412	**27,232.49181**	32,279.82	31,625.92
athlete17	3113	2336	**61,730.61292**	**61,730.61292**	84,420.1	83,286.37
athlete18	2860	2170	**35,525.36677**	**35,525.36677**	40,719.77	41,091.19
athlete19	2893	1840	**98,796.99767**	**98,796.99767**	104,757.72	106,278.73

**Table 4 sensors-24-01700-t004:** Comparison between the proposed model, employing the NM optimization method, and baseline approaches $B_{1}$ (mean) and $B_{2}$ (median), across standard regression metrics (MAE, RMSE, MSE). The two columns (μ and σ) below each metric indicate the mean and standard deviation of the metric values across all players in the dataset. Values printed in boldface represent the superior performance across the columns.

	MAE	RMSE	MSE
	μ σ	μ σ	μ σ
**Model**	**396.87** **61.42**	**520.69** **88.66**	**278,565.18** **97,184.8**
$B_{1}$	429.0484.87	581.34185.84	370,678.57316,947.31
$B_{2}$	421.5795.96	613.47300.11	461,671.73684,901.29

## Data Availability

The source code and all the data used for model fitting are available at the public repository: https://github.com/askoki/expectation-score-profiler (accessed on 28 February 2024).

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
