# Peer review of "Building Individual Player Performance Profiles According to Pre-Game Expectations and Goal Difference in Soccer"

_sensors, 2024, doi:10.3390/s24051700_

Round 1

Reviewer 1 Report

Comments and Suggestions for Authors

This original research effort brings forth a clever method for increasing the “granularity” of team performance that has the potential to be of interest to teams seeking marginal improvements in individual player performance.  Indeed, interest in this study was based on the hope the method would be applied to multiple qualitative factors of player performance, but the analytic scheme does not appear to admit such an analysis at this point. This becomes clear at the end of the paper, Lines 350-354. The literature search is well done, leading to a very informative Table 1, though one seems to get a mixed message,and presumably this is partially the impetus for the present work.  The data acquisition, preprocessing and energy expenditure model are well described, though it really takes several readings to distill the essence of the unique analytic approach. There are no clear flaws with the model fitting, though it leads to the question of whether the aggregate set of assumptions buried within the energy model are possibly equal in magnitude to the results discussed in Section 3. The abundance of data presented in Tables 3-5 is commendable but slightly challenging to set back and grasp full significance. Figure 7 appears to attend to this concern. Figure 9 is a clear outcome of a lot of work, but the table is so hard to read that, again, for the readers seeking to know the “take away” from the work this may not inform the way intended, thought, indeed, the discussion makes the necessary points. In the end the overall impression is a well-done, unique work that takes a lot of digging in and the reader must be vested in achieving marginal player performance.

Author Response

POINT1: This original research effort brings forth a clever method for increasing the “granularity” of team performance that has the potential to be of interest to teams seeking marginal improvements in individual player performance.  Indeed, interest in this study was based on the hope the method would be applied to multiple qualitative factors of player performance, but the analytic scheme does not appear to admit such an analysis at this point. This becomes clear at the end of the paper, Lines 350-354. The literature search is well done, leading to a very informative Table 1, though one seems to get a mixed message,and presumably this is partially the impetus for the present work.  

RESPONSE1: We appreciate the Reviewer's engagement with our manuscript and the insightful comments regarding the analytics scheme and the clarity of Table 1. The Introduction aims to explain different perspectives on how contextual factors, like goal difference, impact players' efforts. Specifically, we highlight conflicting conclusions from various studies on this matter and a need for a more individual approach. In response to the Reviewer's concerns, we've included more details about related work in measuring players' internal and external load, which are important for our analysis.

Admittedly, our current approach has limitations, as pointed out by the Reviewer. However, we view this as a constructive step forward. By acknowledging the current constraints, we are better positioned to refine and enhance our methodology in future iterations.

POINT2: The data acquisition, preprocessing and energy expenditure model are well described, though it really takes several readings to distill the essence of the unique analytic approach. 

RESPONSE2: We appreciate the Reviewer's comment. We recognize that preprocessing and model fitting can present challenges in understanding. To address this, we have restructured the Data Acquisition section into three distinct parts: Study Design, Subjects, and Data Acquisition. This restructuring aims to provide a clearer explanation of the available dataset, which in turn will facilitate comprehension of the subsequent methodology section.

POINT3: There are no clear flaws with the model fitting, though it leads to the question of whether the aggregate set of assumptions buried within the energy model are possibly equal in magnitude to the results discussed in Section 3. The abundance of data presented in Tables 3-5 is commendable but slightly challenging to set back and grasp full significance. Figure 7 appears to attend to this concern. 

RESPONSE3: We appreciate the Reviewer's comment regarding the potential magnitude of errors within our model assumptions. We acknowledge the inherent imperfections in our model and recognize that aggregate assumptions may introduce errors in the results. However, we demonstrate that our approach outperforms current methods used to assess players' efforts, which rely on mean or median metric values across specific goal differences. In response to the concern about the comprehensibility of the extensive data presented in Tables 3-5, we have relocated Tables 4 and 5 to the supplement for those interested in further inspecting the detailed results. Additionally, we have introduced a new Table 4 to directly address the mentioned issue. Table 4 now includes mean squared error, root mean squared error, and mean absolute error values for both our approach and the baseline methods. This addition provides readers with a clear and concise comparison, allowing them to easily grasp the improvements our approach brings to the current state-of-the-art.

POINT4: Figure 9 is a clear outcome of a lot of work, but the table is so hard to read that, again, for the readers seeking to know the “take away” from the work this may not inform the way intended, thought, indeed, the discussion makes the necessary points. In the end the overall impression is a well-done, unique work that takes a lot of digging in and the reader must be vested in achieving marginal player performance.

RESPONSE4: We appreciate the Reviewer for bringing this to our attention. In response, we have relocated the current Fig. 9 to the Supplementary material and introduced a more comprehensible Figure in its place. This new Figure presents the performance of the top five players, ranked by minutes played, across their first ten games. By providing a clearer visualization, we aim to enhance the reader's understanding of the model's performance throughout the matches. The original image remains accessible in the Supplementary material for those interested in a more detailed examination of the data.

Reviewer 2 Report

Comments and Suggestions for Authors

Dear Authors,

In this manuscript, Authors propose a mathematical approach utilizing optimization methods that consider individual player traits and specific distributions, departing from the prevalent group-centric approach in current research. The manuscript focused on the assessment of sports performance using the FIFA approved GPexe pro2 devices. The study will be of interest to many specialists for improving sports performance. However, the manuscript needs to be edited to improve its relevance and the validity of the study. Authors should revise the manuscript to improve its structure and readability. The Authors should also provide the scientific hypothesis in order to construct the study design more correctly.

The Introduction should focus more on scientific approaches to the assessment of sports performance. The information in lines 28-30 is not supported by references.

Authors should be more careful with their references. As can be seen from line 51, references were made to only two studies. This is not “many” as stated.

The section “Material and methods” should be structured. It is better to present information about the subjects (group characteristics, number of subjects, number of measurements, etc.) earlier. All playing positions (5 center backs (CBs), 5 MFs, 5 FWs, 3 wide forwards (WFs), and 1 wing back (WB)) should be explained.

Data Acquisition should be more clearly presented. The wearable activity logging device was not properly specified. The resulting parameters are also not indicated. It should be clearly explained what parameters of the device were used to measure energy expenditure.

The study points/periods should be also explained. As seen in the manuscript, goal difference was assessed retrospectively. The assessment of players' pre-match expectations is not explained.

It is not clear from the study whether this approach was successful. The authors should present the outcome of the study more clearly. Since no statistics were presented, the results were not proven.

As seen from conclusion, “The primary objective of our approach is to objectively quantify players’ mental and physical states, offering valuable insights into their performance”. However, no indicators of mental and physical states of players were presented in the study. Thus, the conclusions are not supported with the results.

The abstract should be edited to better suit the research conducted. Avoid information that is far from the area of study. The parameters that characterize the results obtained should be presented more clearly.

Author Response

POINT1: Dear Authors,

In this manuscript, Authors propose a mathematical approach utilizing optimization methods that consider individual player traits and specific distributions, departing from the prevalent group-centric approach in current research. The manuscript focused on the assessment of sports performance using the FIFA approved GPexe pro2 devices. The study will be of interest to many specialists for improving sports performance. However, the manuscript needs to be edited to improve its relevance and the validity of the study. Authors should revise the manuscript to improve its structure and readability. The Authors should also provide the scientific hypothesis in order to construct the study design more correctly.

The Introduction should focus more on scientific approaches to the assessment of sports performance. 

RESPONSE1: We appreciate the Reviewer for the insightful comments provided. In response, we have restructured the Introduction section into two distinct parts: Related Work and Contributions and Structure. The Related Work segment has been bolstered with recent research on the measurement of external and internal load in soccer players, which serves as a foundational aspect for the studies outlined in Table 1. With this revised structure, we aim to clarify the existing challenges in current research while clearly articulating our objectives and contributions. We believe that this restructuring will enhance the relevance and clarity of our manuscript, thereby improving its readability and validity.

POINT2: The information in lines 28-30 is not supported by references.

Authors should be more careful with their references. As can be seen from line 51, references were made to only two studies. This is not “many” as stated.

RESPONSE2: We appreciate the Reviewer for highlighting the missing references and the need for more thorough referencing in our manuscript. We have addressed the absence of supporting references for the information provided in lines 28-30 and have ensured that all assertions are properly substantiated throughout the text. Moreover, we have revised line 51 to accurately enumerate all the papers cited in Table 1, aligning with our original intention.

POINT3: The section “Material and methods” should be structured. It is better to present information about the subjects (group characteristics, number of subjects, number of measurements, etc.) earlier. All playing positions (5 center backs (CBs), 5 MFs, 5 FWs, 3 wide forwards (WFs), and 1 wing back (WB)) should be explained.

Data Acquisition should be more clearly presented. The wearable activity logging device was not properly specified. The resulting parameters are also not indicated. It should be clearly explained what parameters of the device were used to measure energy expenditure.

The study points/periods should be also explained. As seen in the manuscript, goal difference was assessed retrospectively. The assessment of players' pre-match expectations is not explained.

RESPONSE3: We appreciate the reviewer's insightful comments. In response, we have restructured the "Data Acquisition" section into three distinct parts: "Study Design," "Subjects," and "Data Acquisition." This reorganization aims to enhance clarity and readability. Furthermore, we have specified the playing position groups to provide readers with a clear understanding of each position's role on the pitch. The wearable device and its parameters for measuring energy expenditure have been more thoroughly explained, along with the data capture process. Additionally, we have provided detailed explanations of the goal difference collection process and the method for assessing players' pre-game expectations. These enhancements aim to address the reviewer's concerns and improve the overall comprehensibility of the manuscript.

POINT4: It is not clear from the study whether this approach was successful. The authors should present the outcome of the study more clearly. Since no statistics were presented, the results were not proven.

RESPONSE4: We appreciate the Reviewers' concerns. While we have presented the improvement of our approach compared to the baseline approaches, which utilize mean and median values according to goal difference in Tables 3-5, we understand that this information might be challenging for readers to grasp. Therefore, we have taken steps to enhance clarity.

Tables 4 and 5 have been relocated to the Supplementary material, and a new Table 4 has been introduced. This new table presents widely used regression metrics to assess the validity of the model. Specifically, Table 4 includes mean squared error, root mean squared error, and mean absolute error values for both our approach and the baseline methods, along with mean and standard deviation values across all players in the study.

We believe that these adjustments will facilitate a better understanding of the model's performance and its comparison to baseline methods.

POINT5: As seen from conclusion, “The primary objective of our approach is to objectively quantify players’ mental and physical states, offering valuable insights into their performance”. However, no indicators of mental and physical states of players were presented in the study. Thus, the conclusions are not supported with the results.

RESPONSE5: We appreciate the reviewer for highlighting this discrepancy. We intended to emphasize the mental shift in players' behavior (effort) when considering expectations before the game and goal difference states. However, we recognize that the phrasing may have implied different mental profiles and psychological approaches, which was not the focus of our study. To address this, we have revised the sentence in question and updated it in the abstract as well. Additionally, we have enhanced the validity of our approach by introducing the new Table 4, as previously mentioned.

POINT6: The abstract should be edited to better suit the research conducted. Avoid information that is far from the area of study. The parameters that characterize the results obtained should be presented more clearly.

RESPONSE6: We appreciate the reviewer's feedback regarding the abstract. In response, we have thoroughly revised and updated the abstract to ensure that it accurately reflects the focus of our research. Specifically, we have incorporated the key metrics presented in Table 4 to provide a clearer summary of our results, making it easier for readers to understand the outcomes of our study.

Round 2

Reviewer 2 Report

Comments and Suggestions for Authors

In this manuscript, Authors propose a mathematical approach utilizing optimization methods that consider individual player traits and specific distributions, departing from the prevalent group-centric approach in current research. By employing two optimization algorithms, Authors evaluate the reliability of the approach, pinpointing potential areas of weakness. The manuscript has been improved in its structure. Readability has improved. However, some editing of the English language is necessary to ensure uniform terminology and vocabulary.

Comments on the Quality of English Language

some editing of the English language is necessary to ensure uniform terminology and vocabulary.

Author Response

POINT1: In this manuscript, Authors propose a mathematical approach utilizing optimization methods that consider individual player traits and specific distributions, departing from the prevalent group-centric approach in current research. By employing two optimization algorithms, Authors evaluate the reliability of the approach, pinpointing potential areas of weakness. The manuscript has been improved in its structure. Readability has improved. However, some editing of the English language is necessary to ensure uniform terminology and vocabulary.

RESPONSE1: We appreciate the reviewer’s comments. In response, we have carefully revised the manuscript, both in terms of English language and style, with particular emphasis on the Methodology and Results sections, to enhance the uniformity of terminology and vocabulary. We believe these revisions have further improved the clarity and coherence of our work.